# The Regional Burden and Disability-Adjusted Life Years of Knee Osteoarthritis in Kazakhstan 2014–2020

**DOI:** 10.3390/biomedicines11010216

**Published:** 2023-01-14

**Authors:** Gulnur Zhakhina, Arnur Gusmanov, Yesbolat Sakko, Sauran Yerdessov, Yuliya Semenova, Dina Saginova, Arman Batpen, Abduzhappar Gaipov

**Affiliations:** 1Department of Medicine, Nazarbayev University School of Medicine, Kerey and Zhanibek Street 5/1, Astana Z05P3Y4, Kazakhstan; 2National Scientific Center of Traumatology and Orthopedics Named after Academician Batpenov N.D., Astana Z05P3Y4, Kazakhstan; 3Clinical Academic Department of Internal Medicine, CF “University Medical Center”, Astana Z05P3Y4, Kazakhstan

**Keywords:** knee osteoarthritis, gonarthrosis, burden of disease, disability-adjusted life years

## Abstract

A Global Burden of Disease (GBD) study reported that 9.6 million years lived with disability (YLDs) were lost due to hip and knee osteoarthritis (KOA) in 2017. Although the GBD study presents the disease burden at the global level, there is no information on any Central Asian country. This study aims to investigate the epidemiology of knee osteoarthritis in Kazakhstan. The data of 56,895 people with KOA between 2014–2020 was derived from the Unified National Electronic Health System of Kazakhstan and retrospectively analyzed. The majority of the cohort (76%) were women, of Kazakh ethnicity (66%), and older than 50 years of age (87%). The risk of gonarthrosis escalated for women after 50 years and peaked at 75 years with a rate of 3062 females admitted to hospital per 100,000 women in the population. This observation is approximately three times higher than for men of the same age group. A geographical analysis showed that the Jambyl oblast, West Kazakhstan, North Kazakhstan, and the Akmola oblast have the highest burden of disease. During the observation period, 127,077 age-adjusted YLDs were lost due to knee osteoarthritis. This is the first study in Kazakhstan to investigate the burden of knee osteoarthritis. This research recognizes age and sex-based differences, and regional disparities in the incidence of knee osteoarthritis. This knowledge can lead to the development of more specific diagnostic approaches and gender-personalized therapy protocols for patients.

## 1. Introduction

Knee osteoarthritis (KOA), also called gonarthrosis, refers to orthopedic disorders that affect the musculoskeletal system. The major structural changes in the joints can cause pain and physical disability, affect mental health, and increase the risk of chronic diseases, which lowers the quality of life [1]. The highest burden of disease is observed in the elderly population [2] and females rather than males [3]. Moreover, there are other KOA risk factors such as obesity, knee injury, muscle weakness, and joint laxity [4,5].

According to the Global Burden of Disease (GBD) study, in 2017 there were 303.1 million prevalent cases of hip and knee osteoarthritis (OA) worldwide [6]. The World Health Organization (WHO) uses disability-adjusted life years (DALYs) as a metric to estimate the burden of disease, which sums the years of life lost due to premature death (YLLs) and years lived with disability (YLDs) into one index [7]. The GBD study evaluated that approximately 9.6 million YLDs were lost due to hip and knee OA worldwide in 2017 [6]. The economic consequences of OA have remarkable effects on disease burden, including treatment costs, and loss of productivity of patients and their caregivers [4].

There are no large-scale, population-wide epidemiological data on the KOA burden in Eurasian countries. Although the GBD study presents the disease burden at the global level, there is no information on any Central Asian countries. Therefore, this study aims to evaluate the burden of knee OA with disability-adjusted life years (DALYs) in Kazakhstan, the largest Central Asian country, and to fill the gaps in the literature. Given the economic and health burden of KOA and the fact that the epidemiological parameters of the Kazakhstani population have never been properly assessed, it is clear that accurate estimates are needed.

## 2. Methods

### 2.1. Study Design and Population

The data for this retrospective study was extracted from the Unified National Electronic Health System (UNEHS) of Kazakhstan for 2014–2020. The detailed description of UNEHS databases is given elsewhere [8]. According to the International Classification of Diseases (ICD), knee OA is coded as M17. The raw data of 107,384 hospital admissions during the study period was registered. After data cleaning and management, 56,895 unique population registry numbers (RPN ID) were kept for further analysis. The rate of population growth in Kazakhstan and its regions was obtained from the Statistics Committee under the Ministry of National Economy of the Republic of Kazakhstan [9].

### 2.2. Exposures and Covariates

The data on date of birth, gender, address, ICD-10 diagnosis, date(s) of admission, social status, and death date were acquired if applicable. The birth and death dates were derived from the population registry through RPN ID. The end day of the follow-up period was 31 December 2020. The address information was used to divide the living area into urban and rural, and also by administrative units. There were more than 120 nationalities; therefore, ethnicity was divided into Kazakhs (n = 37,658), Russians (n = 11,872), and others (n = 7365). The information on comorbidities such as obesity, hypertension [10] and diabetes [11] was derived from UNEHS and merged with the database of KOA using RPN IDs.

### 2.3. Disability-Adjusted Life Years Calculation

The calculation of DALYs without age-weighting and time discounting was based on the World Health Organization (WHO) methods for the GBD estimates [7]. The overall YLL was calculated as the sum of years of life lost per age category group (starting from birth to 85+ y.o.). The number of deaths in each group was multiplied by the life expectancy at the age at which the death occurred. According to the GBD 2010 reference life table, the life expectancy at birth for both males and females was 86.0 years [7].

WHO and GBD 2010 use a prevalence-based approach to calculate YLD. The formula of YLD is the multiplication of the prevalence and disability weight (DW). Knee osteoarthritis has different DWs depending on the level of disease severity. Mild intensity, which is described as having some difficulty in running and walking, has a DW of 0.023 [12]. Moderate severity, which is where a person limps and along with the above-mentioned symptoms has sleeping problems due to the pain, has a DW of 0.079 [12]. A severe form of the disease has a DW of 0.165, and a person would have acute pain in the leg and major problems with physical functioning [12]. Unfortunately, the database does not have information on the severity of the pain; therefore, the average of DWs (0.089) was used for the calculation of YLDs in this study.

### 2.4. Statistical Analysis

The incidence of knee osteoarthritis based on hospital admission and discharge records were presented as absolute numbers and per million population (PMP) in each administrative division of Kazakhstan. Data cleaning, data management, and statistical analysis were performed using STATA 16.1 MP2 version (STATA Corporation LLC, 4905 Lakeway Dr, College Station, TX 77845, USA.). The map of knee osteoarthritis incidence rate in 2019 by regions was constructed using QGIS 3.12. It was decided to take data for 2019 due to the possible underestimation of hospital admissions during the COVID-19 pandemic and lockdown restrictions. Moreover, a valid address was defined as a region rather than the location of the hospital where the patient was admitted.

Patients were not involved in the study. Therefore, the requirement for informed consent from study participants was waived by the Nazarbayev University Institutional Review Ethics Committee (NU-IREC 490/18112021). All methods were carried out in accordance with the “Reporting of studies conducted using observational routinely-collected health data” (RECORD) guidelines.

## 3. Results

### 3.1. Socio-Demographic and Baseline Characteristics

The socio-demographic characteristics of the cohort (n = 56,895) are given in Table 1. During 2014–2020, 43,427 (76%) women and 13,468 men (24%) were admitted to the hospital due to knee osteoarthritis. Among the cohort, 49,746 (88%) were older than 50, and 432,550 (57%) were retired. Of the patients, 37,658 (66%) were Kazakhs, 11,872 (21%) were Russian, and the rest were of other ethnicities. Concurrent diabetes mellitus and hypertension were present in 11% and 49% of the cohort, respectively. During the observation period, 7% of the participants died. According to statistical analysis, all socio-demographic factors and comorbidities had statistically significant differences between the genders.

### 3.2. Incidence Based on Hospital Admission Records

The bivariate analysis shows a remarkable difference in the age of males and females at the time of first hospital admission, *p* < 0.001 (Table 1). Age and sex-specific rates of knee osteoarthritis show that females are more predisposed to the disease compared to males (Figure 1). The risk of gonarthrosis escalates for women after 50 years and peaks at 75 years, with a rate of 3062 females admitted to hospital per 100,000 women in the population. This observation is approximately three times higher than for men of the same age group. However, the trend changes after 85 years. At the age of 95 years, women had a rate of 239 females admitted to the hospital per 100,000 women in the population, which is three times lower than the observed incidence for males.

The incidence rate based on hospital admission and discharge status increased over the observation period (Figure 2). However, a decline is observed in 2020. The regional differences in knee osteoarthritis incidence in 2019 are presented in Figure 3. The highest burden was observed in Jambyl oblast, West Kazakhstan, North Kazakhstan, and Akmola. The lowest admission rates were in Shymkent, Turkestan, Atyrau, and Mangystau.

### 3.3. DALY

Age and sex-adjusted YLLs, YLDs, and DALYs based on hospital admission and discharge status are presented in Table 2. Over the observation period, 202,373 DALYs were lost in Kazakhstan due to knee osteoarthritis. The YLL was 75,296, and the YLD was 127,077. The highest burden of gonarthrosis was observed for people 55–69 years old with an overall burden of 107,479.5 DALYs. The higher contribution of premature death to DALY is observed after 65 years when YLLs are overweight YLDs.

The female-to-male ratio in contribution to DALY is 2.3 to 1. Total DALYs for women is approximately three times larger than for men, 141,364.3 and 61,008.7 DALYs respectively. Although the gender tendency is clear, the burden of disease is higher for males than for females in the age group of 0 to 39 years.

## 4. Discussion

This is the first study in Kazakhstan that evaluated the burden of knee osteoarthritis based on hospitalization records between 2014 and 2020. In the cohort, only a quarter of the patients were males, the majority were Kazakhs, older than 50, and they lived in an urban area. Approximately half of the patients had hypertension as a comorbid condition. Although the incidence rate per million population increased over the observation period, there was a decline in 2020. Moreover, the highest burden of disease was observed in the Jambyl oblast; however, the nearby administrative units as Turkestan and Shymkent showed the lowest incidence rate of knee osteoarthritis. In spite of the fact that females had a higher contribution to disability-adjusted life years, gonarthrosis was more burdensome for males of age younger than 40 years.

### 4.1. Socio-Demographic and Baseline Characteristics

In Kazakhstan, the majority of the admitted patients for hospitalization were women during the observation period. The higher frequency of knee osteoarthritis in females is recorded in the literature [13,14,15]. Moreover, studies show that the incidence of gonarthrosis escalates in women after approximately 50 years of age [16,17], which is consistent with the results of the current study. Although scientists and clinicians tried to explain the burden of disease in women with estrogen deficiency after menopausal age [18,19], some research shows that this theory/association is conflicting [20]. The systematic review of knee osteoarthritis highlights a significant variance in joint morphometry, kinematics, and pain severity depending on gender [21]. Along with the biological differences between males and females, it is possible that women tend to use healthcare services more frequently than men [22,23], which explains the considerably lower proportion of the latter in a cohort. The results of this study can contribute to the development of gender-specific screening approaches and interventions to reduce the burden of knee osteoarthritis in Kazakhstan.

In addition to the female gender, knee osteoarthritis is highly correlated with aging. The results of this study show that the majority of patients are older than 50, and more than half are retired. Structural and cellular changes in the cartilage and bones that escalate with higher age are linked to the pathogenesis of osteoarthritis [24]. Although the literature shows that gonarthrosis is more prevalent in rural areas [25,26], in this study, the majority were from urban settlements. Due to the fact that the cohort of current research is formed from the hospital admission and discharge records, there is the possibility of actual disease prevalence underestimation in Kazakhstan. Moreover, unequal access to healthcare and poorly developed health infrastructure in rural areas can cause barriers to visiting physicians and surgeons [27].

Although KOA is more frequent in females, according to the age and sex-specific incidence of disease, men of working age, the principal sector in society that contributes to Kazakhstan’s economy, carry a higher burden than women. The incidence of KOA in females is lower than in males up to 45 years of age, and then skyrockets after this cut-off point. A similar result was observed in the Rotterdam Study cohort, and lower education level was one of the predictors of a higher risk of KOA in men [28]. In addition, this difference can be associated with occupational risks that females and males face at work. According to the employment and labor statistics of Kazakhstan, mostly men worked in the agriculture, forestry, industry, and construction sectors [29]. Masons and construction workers, people working in the agriculture sector, and farms have elevated risks of OA due to heavy labor and constant stress on specific joints [30]. Prevention of KOA in young adults can be enhanced by reducing the intensity of physical labor in these sectors. In addition, educational campaigns on the importance of maintaining a healthy weight, healthy diet, and avoiding smoking can increase the awareness of people in these workplaces. Furthermore, employers must provide employees with the appropriate equipment to work with heavy weights to minimize the pressure on the musculoskeletal system.

According to the results of this study, approximately half of the participants had been diagnosed with hypertension. The relationship between high blood pressure and osteoarthritis is still controversial. Some research shows that there is a causal association between them [31]; however, other research claims that hypertension has only confounding effects [32]. On the other hand, a minority of the cohort had diabetes as a comorbid condition. Although diabetes is not a risk factor for gonarthrosis, researchers show that it has an effect on the higher severity of pain and on the location of pain [33,34].

### 4.2. Incidence Based on Hospital Admission and Discharge Records

In Kazakhstan, the incidence rate of knee osteoarthritis based on hospital admission and discharge status increased over the observation period. However, there was a decline in the rate in 2020. Due to the COVID-19 pandemic that escalated in 2020, major medical associations recommended postponing elective surgical procedures to minimize the risk of infection spread [35]. Moreover, the National Health Service and the World Health Organization advised repurposing the available resources from not urgent conditions, such as orthopedic diseases, to support the influx of COVID-19 cases [36,37]. Garrido-Cumbrera and colleagues investigated the impact of the pandemic on patients with musculoskeletal diseases and found that disruption in access to healthcare services and elective surgeries led to the decline of mental health and quality of life [38]. The impact of the inadequacy of the healthcare resources for osteoarthritis patients during the pandemic and the following consequences on the socio-economics of the country can be investigated separately.

The results of this study show some geographical disparities in knee osteoarthritis incidence in Kazakhstan. There is no trend in regional propensity/emplacement of administrative units with the highest rates of gonarthrosis (Jambyl oblast, West Kazakhstan, North Kazakhstan, and Akmola oblast). According to the demographic data of Kazakhstan for 2019, these regions had a high ratio of people older than working age [9], also known as retirees, which is one of the main predictors of knee osteoarthritis. In addition, studies from Asian countries show that people who have farming, agriculture, and fishing as their main occupations tend to have an increased risk of osteoarthritis [26,39,40]. According to employment statistics of Kazakhstan by type of economic activity in 2019, the Akmola, Jambyl, and North Kazakhstan regions had higher proportions of agriculture, forestry, and fisheries as an occupation [41]. These types of employment can lead to traumatic events and can involve repetitive stress on joints.

### 4.3. DALY

The Global Burden of Disease study in 2016 defined osteoarthritis as the 12th leading cause of disability [42]. Therefore, years lived with a disability is the metric used to present the disease burden [43]. The results of this study are consistent with the literature, and the contribution of YLDs to DALY was higher. In addition, the cohort studies show that hip or knee osteoarthritis was associated with a higher risk of all-cause and cardiovascular mortality [44,45,46]. The age and sex-adjusted DALY analysis illustrate that YLLs outweigh YLDs after 70 years of age.

### 4.4. Strengths and Limitations

This research has several advantages. First of all, knee osteoarthritis epidemiology and its burden in Kazakhstan were evaluated. There were no large studies evaluating the loss of disability-adjusted life years related to gonarthrosis in the country, even in Central Asia. The findings of our study may potentially extrapolate to other Central Asian countries, taking into account our common culture, lifestyle, and healthcare model. The finding can help to evaluate the economic effect of disease and develop preventative measures and better treatment protocols. Moreover, the research demonstrates geographical disparities in the incidence of knee osteoarthritis. This knowledge can lead to changes in health policies and provide direction to health policymakers.

Despite the mentioned advantages, there are some limitations to this study. Firstly, the evaluation of disease epidemiology is solely based on hospital admission and discharge records, which do not represent all knee osteoarthritis cases in the country. The database lacks information on important risk factors such as dietary habits, previous joint injuries, and family history of OA. The information on obesity was linked from reported data, and it has the potential to underestimate real numbers in the cohort. Moreover, the disability weight used for the estimation of DALY is an average of DWs for three severity levels. There is no information on pain intensity, which would allow us to make a more precise calculation.

## 5. Conclusions

This is the first study in Kazakhstan, even in Central Asia, to evaluate the epidemiology of knee osteoarthritis and to present the burden in terms of disability-adjusted life years. The retrospective analysis was based on hospital admission and discharge records of patients all over the country between 2014 and 2020. The results show a significantly higher proportion of females in the cohort and a higher predisposition to knee osteoarthritis after 50 years of age. Moreover, the analysis shows that the incidence of osteoarthritis is increasing each year. Further educational campaigns aimed at disease prevention and reduction of risk factors are needed. In addition, the effect of separate comorbid conditions, especially obesity, on the progression of KOA can be evaluated in future research. The assessment of the cost-effectiveness of knee replacement, rehabilitation of patients after surgery, and its effect on the quality of life can be studied separately.

## Figures and Tables

**Figure 1 biomedicines-11-00216-f001:**
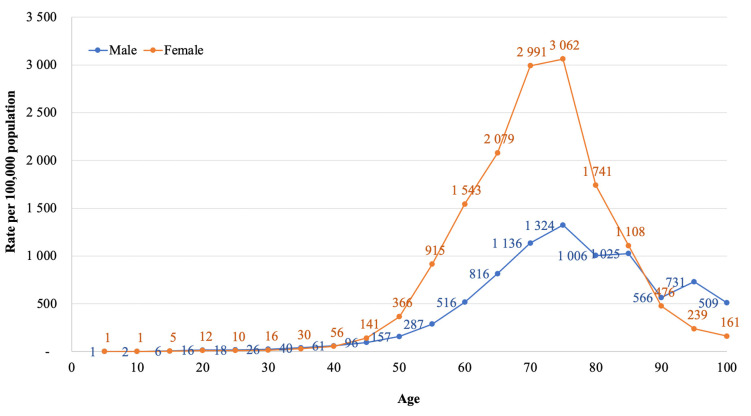
Age and sex-specific rates of knee osteoarthritis per 100,000 population in Kazakhstan from 2014–2020.

**Figure 2 biomedicines-11-00216-f002:**
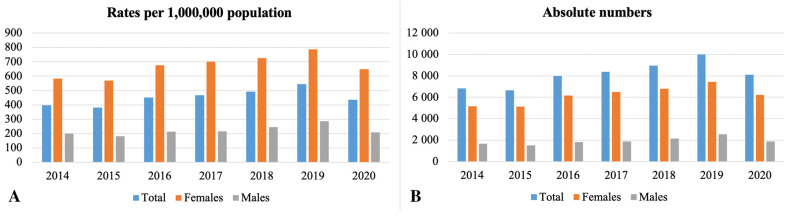
The burden of knee osteoarthritis in Kazakhstan by years based on admission and discharge status: (**A**) incidence (overall, females, males) per 1,000,000 population; (**B**) absolute numbers (overall, females, males).

**Figure 3 biomedicines-11-00216-f003:**
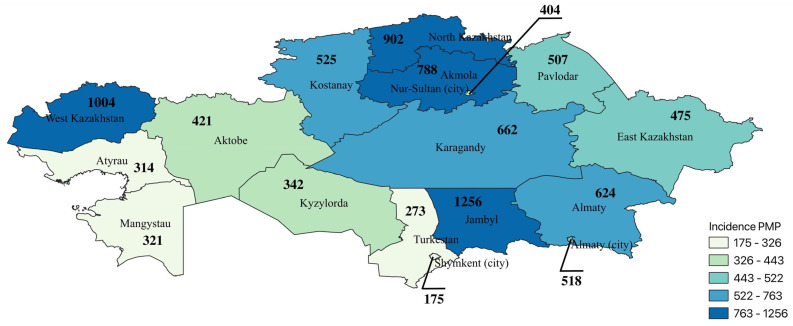
Incidence of knee osteoarthritis hospital admission in Kazakhstan in 2019 by regions.

**Table 1 biomedicines-11-00216-t001:** Socio-demographic and medical characteristics of patients with knee osteoarthritis for the years 2014–2020.

	Total (n = 56,895)	Female (n = 43,427; 76%)	Male (n = 13,468; 24%)	*p*-Value
Socio-Demographics				
Age category, n (%)				<0.001
<17 y.o.	193 (0.3)	77 (0.2)	116 (0.9)	
18–34 y.o.	1105 (2)	446 (1)	659 (5)	
35–50 y.o	5851 (10)	3931 (9)	1920 (14)	
51–69 y.o.	36,898 (64.7)	28,968 (66.8)	7930 (59)	
>70 y.o.	12,848 (23)	10,005 (23)	2843 (21.1)	
Ethnicity, n (%)				<0.001
Kazakh	37,658 (66)	27,977 (64)	9681 (72)	
Russian	11,872 (21)	9657 (22)	2215 (16)	
Other	7365 (13)	5793 (14)	1572 (12)	
Living area, n (%)				<0.001
Urban	34,079 (60)	26,838 (62)	7241 (54)	
Rural	22,816 (40)	16,589 (38)	6227 (46)	
Social status, n (%)				<0.001
Employed	11,761 (21)	7842 (18)	3919 (29)	
Unemployed	8607 (15)	6099 (14)	2508 (19)	
Retiree	32,550 (57)	27,051 (62)	5499 (41)	
Disabled	1584 (3)	905 (2)	679 (5)	
Other	2393 (4)	1530 (4)	863 (6)	
Outcome				<0.001
Living	53,108 (93)	41,068 (95)	12,040 (89)	
Died	3787 (7)	2359 (5)	1428 (11)	
Comorbidities				
Obesity, n (%)				<0.001
Yes	1090 (2)	888 (2)	202 (1)	
No	55,805 (98)	42,539 (98)	13,266 (99)	
Hypertension, n (%)				<0.001
Yes	27,897 (49)	22,674 (52)	5223 (39)	
No	28,998 (51)	20,753 (48)	8245 (61)	
Diabetes, n (%)				<0.001
Yes	6398 (11)	5248 (12)	1150 (9)	
No	50,497 (89)	38,179 (88)	12,318 (91)	

**Table 2 biomedicines-11-00216-t002:** Age and sex-adjusted years of life lost due to premature death (YLLs) and years lived with disability (YLDs), and disability-adjusted life years (DALYs) for knee osteoarthritis in Kazakhstan from 2014–2020.

Age Group	Life Expectancy	Female			Male			Total		
		YLL	YLD	DALY	YLL	YLD	DALY	YLL	YLD	DALY
0-19	76.2525	0.0	687.6	687.6	0.0	1022.1	1022.1	0.0	1709.7	1709.7
20-24	63.88	63.9	392.5	456.4	127.8	728.0	855.8	191.6	1120.6	1312.2
25-29	58.94	117.9	656.1	774.0	235.8	1044.8	1280.5	353.6	1700.9	2054.6
30-34	54	216.0	1029.7	1245.7	324.0	1328.7	1652.7	540.0	2358.5	2898.5
35-39	49.09	392.7	1500.0	1892.7	490.9	1571.2	2062.1	883.6	3071.2	3954.8
40-44	44.23	442.3	3238.6	3680.9	309.6	2060.7	2370.3	751.9	5299.3	6051.2
45-49	39.43	1104.0	6974.3	8078.3	1025.2	2691.6	3716.7	2129.2	9665.8	11,795.1
50-54	34.72	3888.6	15,305.0	19,193.6	2916.5	4051.8	6968.3	6805.1	19,356.8	26,161.9
55-59	30.1	6110.3	20,409.4	26,519.7	4605.3	5622.1	10,227.4	10,715.6	26,031.4	36,747.0
60-64	25.55	6821.9	18,161.5	24,983.4	5340.0	5003.2	10,343.2	12,161.8	23,164.7	35,326.5
65-69	21.12	10,792.3	15,391.7	26,184.0	5617.9	3604.0	9221.9	16,410.2	18,995.6	35,405.9
70-74	16.78	7450.3	7451.6	14,901.9	3557.4	1734.7	5292.0	11,007.7	9186.3	20,194.0
75-79	12.85	6270.8	3398.2	9669.0	3341.0	856.7	4197.7	9611.8	4254.9	13,866.7
80-84	9.34	1802.6	755.5	2558.1	1158.2	268.1	1426.2	2960.8	1023.6	3984.4
85+	5.05	444.4	94.5	538.9	328.3	43.6	371.8	772.7	138.1	910.7
**Total**		**45,918.1**	**95,446.3**	**141,364.3**	**29,377.6**	**31,631.1**	**61,008.7**	**75,295.7**	**127,077.3**	**202,373.0**

## Data Availability

The data presented in this study are available on request from the corresponding author. The data are not publicly available due to restrictions from the Republican Center for Electronic Health of the Ministry of Health of the Republic of Kazakhstan, the Ministry of Health of the Republic of Kazakhstan.

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
