# Peer review of "The Regional Burden and Disability-Adjusted Life Years of Knee Osteoarthritis in Kazakhstan 2014–2020"

_biomedicines, 2023, doi:10.3390/biomedicines11010216_

Round 1

Reviewer 1 Report

Novel research focusing on the prevalence of gonarthrosis in a Central Asian country.

Were just hospital admissions considered or were in hospital outdoor visits also counted. Generally hospital admissions tend to be more severe.

The calculation of DALYs is explained well.

Obesity is a huge predictor of Gonarthrosis. It would have been interesting to observe how increased BMI could result in faster progression of disease.

Within this cohort of individuals what percentage underwent knee replacement and how it affected their quality of life would be an interesting observation.

Author Response

We would like to express our deepest gratitude for the time and effort you spent on our manuscript. All the outstanding comments enlightened us, and they will ultimately improve the scientific and practical content of our paper. Please find below the responses to your comments:

  1. Were just hospital admissions considered or were in hospital outdoor visits also counted. Generally hospital admissions tend to be more severe.

Yes, only hospital admission records were included for analysis. This limitation is noted in the Discussion section. In addition, we have cited the paper that describes in detail the databases of the Unified National Electronic Health Systemof Kazakhstan, where our retrospective data originates. It is added in the subsection “Study design and population” of Methods, line 2.

  1. Obesity is a huge predictor of Gonarthrosis. It would have been interesting to observe how increased BMI could result in faster progression of disease.

Thanks for the valuable comment. We have added information on obesity in Table 1 (section of comorbidities). The information on obesity was linked from reported data, and it may not show the real numbers of obese people in the cohort. This was included in the limitations section.

The aim of this study was to investigate the burden of KOA in Kazakhstan by presenting its epidemiology and DALYs. Therefore, we did not analyze the effect of separate comorbidities on disease progression. However, we have added this suggestion as the idea the further research in Conclusion section.

  1. Within this cohort of individuals what percentage underwent knee replacement and how it affected their quality of life would be an interesting observation.

As the aim of this study was to evaluate the burden of KOA in terms of DALYs and general epidemiology, we did include information on surgeries. We have added this suggestion as the idea the further research in Conclusion section.

Reviewer 2 Report

Title: The regional burden and disability-adjusted life years of knee osteoarthritis in Kazakhstan 2014-2020

This article seems well built and brings evidence of a phenomenon not yet fully understood and that certainly deserves further study.

Some points of revision are provided below

Abstract

-       I suggest to expanding the abstract ……..with a strong conclusion

Introduction

-       I suggest expanding the introduction section / now is very poorly

Author Response

We would like to express our deepest gratitude for the time and effort you spent on our manuscript. All the outstanding comments enlightened us, and they will ultimately improve the scientific and practical content of our paper. Please find below the responses to your comments:

  1. Abstract:  I suggest to expanding the abstract ……..with a strong conclusion

Thanks for the advice. We took into the consideration this suggestion and expanded the conclusion with stringer statements.

  1. Introduction: I suggest expanding the introduction section / now is very poorly

Noted with thanks. The structure and content of the Introduction section was changed. The revised manuscript is in tracking mode, so you can see the changes.

Reviewer 3 Report

This paper aims to analyze the burden of knee osteoarthritis (KOA) in Kazakhstan based on disability-adjusted life years (DALY) analysis from 2014 to 2020. The data from 56,895 patients with KAO were collected and statistical methodologies were used to analyze the sociodemographic and medical characteristics of KOA. Statistical results showed that, according to DALY analysis, women older than 50, 55-69 in particular suffer significantly more from the KAO than men in the same age range in Kazakhstan. However, the males of 0-39 years old carry a higher burden of KOA than the females do at the same age range. Overall, this article provides important statistical information for Kazakhstan Government to develop better and more specific strategies against patients with KOA. Some suggestions may help improve the scientific soundness of this article.

1. It was incorrect to write ”The pain caused by OA can reduce physical impairment, affect mental health, and increase the risk of chronic diseases” at the beginning of the Introduction in the manuscript. The “impairment” should be replaced with “repairment”?

2. The rate numbers per 100,000 population for males and females exhibited in figure 1 were very difficult to be differentiated, particularly those at ages less than 50. Thus, I suggest that the authors label the numbers with the respectively same colors as those for the curves representing males (in orange) and females (in blue). 

3. While males at ages ranging from 0-39, principal manpower in society to make a significant contribution to the Country, carry a higher burden of KOA than females do in the same age range, the authors should use a paragraph in the Discussion to particularly address how this disadvantage can be corrected and make suggestions for the policy-making institute in Kazakhstan.

Author Response

We would like to express our deepest gratitude for the time and effort you spent on our manuscript. All the outstanding comments enlightened us, and they will ultimately improve the scientific and practical content of our paper. Please find below the responses to your comments:

  1. It was incorrect to write ”The pain caused by OA can reduce physical impairment, affect mental health, and increase the risk of chronic diseases” at the beginning of the Introduction in the manuscript. The “impairment” should be replaced with “repairment”?

Noted with thanks. The sentence was changed to “The major structural changes in the joints can cause pain and physical disability, affect mental health, and increase the risk of chronic diseases, which lowers the quality of life”.

  1. The rate numbers per 100,000 population for males and females exhibited in figure 1 were very difficult to be differentiated, particularly those at ages less than 50. Thus, I suggest that the authors label the numbers with the respectively same colors as those for the curves representing males (in orange) and females (in blue). 

We followed your advice and made suggested changes.

  1. While males at ages ranging from 0-39, principal manpower in society to make a significant contribution to the Country, carry a higher burden of KOA than females do in the same age range, the authors should use a paragraph in the Discussion to particularly address how this disadvantage can be corrected and make suggestions for the policy-making institute in Kazakhstan.

Thanks for the comment. This issue was considered as a separate paragraph in Discussion section (Socio-demographic and baseline characteristics, paragraph 3).

Round 2

Reviewer 3 Report

After the authors properly addressed my concerns, I have no further questions.